# Taste Panellists’ Evaluations in Official Cheese Competitions: Analysis for Improvement Proposals

**DOI:** 10.3390/foods13233769

**Published:** 2024-11-25

**Authors:** Patricia Hernández-Arencibia, Pedro Saavedra, Conrado Carrascosa Iruzubieta, Elizardo Monzón, Esther Sanjuán

**Affiliations:** 1Nutrition and Bromatology Unit, Department of Animal Pathology and Animal Production, Bromatology and Food Technology, Faculty of Veterinary, Universidad de Las Palmas de Gran Canaria, 35413 Arucas, Las Palmas, Spain; patricia.hernandez119@alu.ulpgc.es (P.H.-A.); esther.sanjuan@ulpgc.es (E.S.); 2Department of Mathematics, Universidad de Las Palmas de Gran Canaria, Mathematics Building, Campus Universitario de Tafira, 35018 Las Palmas de Gran Canaria, Las Palmas, Spain; pedro.saavedra@ulpgc.es; 3Agrarian Area and Agricultural-Livestock and Fisheries Development Service, Cabildo de Gran Canaria, North General Road, 35413 Arucas, Las Palmas, Spain; emonzong@grancanaria.com

**Keywords:** cheese, sensory evaluation, official competition, tasting panel, tasters’ agreement

## Abstract

Sensory analysis is a tool for determining cheese quality by tasting during official competitions, which are useful for revitalising the local cheese sector. This work aims to acquire information about the outcomes of official cheese tastings on Gran Canaria Island (Spain) and analyse this information to improve the sampling methodology, as a possible reference for similar events held elsewhere worldwide. The results of four consecutive tasting competitions were studied over 4 years. The annual scores for odour, taste, texture and overall impression, given by 26 taste panellists (5 permanent), were analysed. This gave 2291 evaluations of 329 cheeses from 13 different varieties. A mixed model was applied with year and cheese variety as fixed effects, and taster and cheese as random effects. Agreement among the permanent tasters’ scores was considered by the intraclass correlation coefficient. The results indicated significant differences in the final scores according to the considered year and cheese variety and suggested a lack of stable patterns initially, but a movement towards homogeneity in the later years. The vegetable coagulant and sheep/goat’s milk semi-matured cheeses obtained the best scores, and the cows’ milk and pasteurised semi-mature cheeses, the worst. All the sensory variables significantly distinguished the cheese varieties, but not texture and taste in the last competition. Agreement among permanent tasters was significant in the last 2 years.

## 1. Introduction

Sensory analysis is a common tool employed to determine the quality of different foods at distinct demand levels, from consumers to expert judges. This analysis can be performed with instruments or by tasters [1], who evaluate products following set criteria [2]. Depending on their knowledge, tasters are classified as experts with high verified sensory sensibility and specific training in products, who are capable of detecting any change; trained judges or panellists, with technical/practical training, who can detect certain food properties and characteristics and evaluate them; laboratory judges, with similar training as trained panellists, but who do simpler tests; and consumers, who are not trained in performing a sensory analysis but know the product or are potential consumers of new products [2,3,4].

To quantify a food sensory analysis, different tests apply that tend to use scales to classify products’ characteristics by means of a magnitude. Depending on the test or characteristics to be evaluated, a verbal, numerical, graphic scale, or a combination of these, is selected. The first two scales are the most widely used, for which linear or categorical scales are employed [5].

Sensory analysis is applied to evaluate a large number of foods, including cheese. Cheese is a basic food eaten worldwide. Fox and McSweeney (2017) describe it as “a group of fermented milk-based food products, produced in a wide range of flavours and forms throughout the world” [6]. According to the United Nations Food and Agriculture Organisation, approximately 26 million tons of cheese made with cow, goat and sheep’s milk were produced in 2020 [7]. What is more, the consumption of this product was about 9.2 million metric tons in 2023 within the European Union, meanwhile in China, it was about 409 thousand metric tons in 2019 [8].

European cheese-making and consumption data indicate that Spain is the seventh-largest cheese-making country in Europe. Its production is 548,410 tons [9] and the inhabitants of this country ate 362,962.760 kg [10] in 2021, with a mean consumption of 7.84 kg/inhabitant. The Canaries Autonomous Community led Spanish cheese consumption with 11.9 kg/habitant. Hence, the importance of this food for consumers in the Canary Islands with production only on the Gran Canaria Island of 1500 tons/year [11]. In addition to being the Spanish region where most cheese is eaten, the cheeses produced in this region are of excellent quality and this territory has obtained many World Cheese Awards since 2003, which is the most important cheese competition in the world. Indeed, in its latest edition (2023), Canary Islands cheeses were awarded 38 medals, namely, 3 super gold medals, 2 gold medals, 13 silver medals and 20 bronze medals [12].

On the Canary Islands, official competitions of this type are also held, and awards are given to the best cheeses in the Canaries Autonomous Community, albeit on a lesser scale. These regional competitions, promoted by public organisations, are used as a marketing tool to promote cheeses, especially artisanal cheeses, that are usually locally distinguished or of Protected Designation of Origin (PDO) [13]. In these cases, the tasters who form part of the tasting process are expert judges, trained tasters, or panellists [14].

To make a sensory evaluation of cheese, it is necessary to bear in mind that cheeses’ organoleptic characteristics depend on many factors, such as raw material, milk’s origin, the cheese-making process, and ripening times [15,16], among others. For instance, cheeses made under the same conditions, but with a mixture of goat and sheep’s milk, have been determined to be softer than those made only with the milk of either of these species [17] for feta cheeses and cheeses like Picante [18]. Likewise, livestock’s food may also have an influence, and cheeses can be made with more intense floral, fruity and plant flavours when their milk comes from animals that are fed in fields as opposed to those fed concentrated feed and forage [16]. pH and fat content are also affected by animals’ diet, and both are higher in animal feed with higher fibre content and also confer cheese more intense aromas and *odours* [19].

Cheese ripening is another parameter with a stronger effect on cheese organoleptic characteristics, and may impact the sensory analysis more than animals’ diet [20]. This parameter affects the intensity of aroma and flavour, and both increase with cheese ripening time [20,21,22,23,24]. Similarly, their persistence over time also grows [24] and a wider variety of aromas, odours and flavours is detected [23]. These changes, triggered by the ripening time, are influenced by factors like moisture content, ripening time, cooking temperature and pH at draining. However, they primarily result from the action of rennet enzymes and the cultured or residual microorganisms in the curd, which initiate primary (lipolysis, proteolysis, and lactose metabolism) and secondary (fatty acid and amino acid metabolism) biochemical processes [25].

To correctly perform a sensory analysis of cheeses, knowing how to distinguish those made of vegetable, animal-based, or commercial rennet is also important. Those made with vegetable coagulant present more intense odours and flavours than those made of commercial rennet, and the commercial kinds are less acidic than those made with mixed animal-based/vegetable coagulants. Regardless of ripening time, cheeses made with vegetable coagulant stand out because their astringency and bitterness are higher, while those made with mixed coagulants present intermediate descriptors between vegetable and animal-based rennet, and their astringency is more like those made with animal-based rennets [26].

Another noteworthy point is that, although the cheeses made according to a single PDO must have similar characteristics because their composition is alike, variations in their texture, colour and some sensorial attributes have been found [21]. This is because it is extremely difficult to standardise cheese making 100% in a single PDO seeing that, as they are artisanal products, each cheesemaker has his or her own way of making cheese, which confers on them an extra cheese sensorial guarantee value [21].

Each taster’s evaluation of all these parameters made by performing a sensory analysis of cheeses generates a vast volume of data, which provides information about their characteristics and how tasters act. To detect any differences among the scores given by a tasting panel, statistical tools are used because they allow the truthfulness of the supplied data to be established [27]. By means of such statistical analyses, it is possible to determine the means and standard deviations of a given product’s sensorial properties and, at the same time, analyse data that could, for example, determine the attributes that stand out the most, to know the effect of ripening time or to compare different cheeses [23,28]. Statistical analysis also allows the correlation, repetitiveness and reproducibility of data obtained from the different tasting panel members to be evaluated [29].

The present work aims to study the factors that could influence the evaluations made by a tasting panel during an official cheese competition and to know their preferences, identify any biases that may arise in the followed methodology and consider what improvements could be made to obtain greater criteria homogeneity in panels of such samplings. The information acquired in this study will be useful for all cheese-tasting competitions anywhere in the world. In addition, it contributes to the academic and scientific fields, since it helps to validate the functional use of sensory analysis methodology, which is market-relevant and examines the specific practices applied in these official competitions.

## 2. Materials and Methods

In order to conduct this study, the data from the annual tastings that were held during the “Official Competition of Cheeses from Gran Canaria” in 2019, 2020, 2021 and 2022 were analysed. Thirteen different cheese varieties were evaluated every year, with ninety cheese samples in 2019, eighty-eight in 2020, 71 in 2021 and eighty in 2022. According to each competition’s rules, any owner of a dairy cheese-making establishment registered with the General Health Register of Food and Feed Companies or with the corresponding regional register can participate [30]. These competitions include artisanal and industrial cheeses.

The *pasteurised* cheese varieties are made using *goat*, *cow,* or *sheep’s* milk, or a mixture of these, and the *soft*, *semi-mature* and *mixed-milk mature* cheeses have non-homogeneous milk proportions among the different cheeses. Ripening times range from 7 and 34 days for *soft* cheeses, 35–105 days for *semi-mature* cheeses and longer than 105 days for *mature* cheeses [30]. The *mixed-coagulants* cheese varieties are made from a mixture of rennet or authorised enzyme and *vegetable coagulant* from cardoon flower *Cynara cardunculus* var. *Ferocísima* or *Cynara scolymus* (with no set proportion), while the *vegetable coagulant* variety is made using 100% *vegetable coagulant* [31].

Each year, the number of taste panellists ranged between 11 and 17, with a minimum number of 5 people per evaluated cheese variety [30]. Each taster has to sample between five and eight different cheese varieties. The distribution of tasters and cheeses in each competition is described in Table 1.

The judges who formed the tasting panel were recruited as ‘external’ based on general criteria (availability, attitude, descriptive knowledge and aptitude), as well as health and psychological criteria (interest, motivation, responsibility, concentration, critical judgment and cooperation). These selected panellists were considered trained judges or laboratory judges, by their training, since they were people from the restaurant/catering sector; other cheese-related groups also participated. In all, 26 tasters took part in the four study years, 329 cheeses were evaluated and 2291 observations were made. Of the 26 tasters, 5 participated every year, and these 5 also sampled the *vegetable coagulant* and *mixed-coagulants* cheese varieties.

To evaluate cheeses, every product was first assigned a random three-digit number (Figure 1) as set out in Standard UNE-EN ISO 6658:2019 [32]. Next, a sample of each variety was cut into a hexahedral shape and placed on a tray (Figure 2). Afterwards, tasting commenced, and four marker descriptors were evaluated: *odour*; *mouth texture*; *Flavour, persisting taste or aftertaste;* and *overall impression.*

The scoring was carried out using a categorical scale from 1 to 7, where 1 was the minimum (“bad”) and 7 was the maximum (“excellent”). For tasters to give each cheese a final score, the weighting factors of 3, 4, 6 and 2 were respectively employed and obtained as follows:Final score=3·Odour+4·Mouth texture+6·Flavour, persisting taste, aftertaste+2·Overall impression

Thus, the score given to each tasted cheese could range from 15 (minimum score) and 105 (maximum score).

These descriptors and their weightings have been employed in these tasting competitions since they started in 1993, for which the following are taken as a reference: the tasting record of the PDO *Queso de Flor de Guía*, *Queso de Media Flor de Guía* and *Queso de Guía* (the only three cheeses from the island protected by a PDO and made with a mixture of milks (sheep/cow/goat of >60%/<40%/<10%) and *vegetable coagulant* exclusively as either a minimum of 50% or an indistinct mixture with animal-based rennet, respectively) [31]. The weighting criteria are set after a consensus is reached by the cheesemakers, tasters and the local administration in charge of organising the tastings. After obtaining the total score, the best median of the evaluations made by tasters is selected to obtain the winning cheese.

The competition organisers decided to not perform the visual phase of the sensory analysis during the competition as this they carried out in advance when deciding whether a cheese was to be included or not in the tasting competition.

Tastings were carried out in a room without noise or smells, but with suitable lighting, in compliance with Standard UNE-EN ISO 8589:2010 [33]. Tables were individual and separated from other tables by an approximate distance of 1 metre to, thus, favour judges’ well-being and not allow them to interact, which removes the influence of personal factors. Tasters were divided into two panels and each subpanel tasted half the varieties. The *mixed-coagulants* and *vegetable coagulant* cheeses were tasted by all the panel members. For palate cleansing, pieces of green apples, spongy bread and light mineral water were used.

The tastings over the four years of the study were consistently conducted between May and June in the mid-afternoon, in accordance with the UNE-ISO 6658:2019 Standard [32].

### 2.1. Statistical Analysis

#### 2.1.1. Mixed-Effects Model to Estimate the Effect of Year and Cheese Variety on Marker Variables

To identify the factors associated with each assessment criteria (*odour*, *texture*, *flavour*, *overall impression* and *final score*), mixed models (MMs) were used, in which the *taster* and *cheese* effects were taken as random effects. As the data exploration suggested that interactions existed between the factors “Year” and “Variety” (scores evolved depending on cheese variety), their effects on each assessment criteria were analysed by separate MMs [34]. More specifically:

***M1. For each “assessment criteria” and variety,*** an MM was estimated as follows (evaluating scores’ evolution for each variety separately):$$y_{i,j,k}=\mu +Year_{k}+Taster_{i}+Cheese_{j}+e_{i,j,k}$$

***M2. For each “assessment criteria” and year,*** an MM was estimated as follows (evaluating within each year the effects that cheese varieties have on scores):$$y_{i,j,k}=\mu +Variety_{k}+Taster_{i}+Cheese_{j}+e_{i,j,k}$$

Here, $y_{i,j,k}$ denotes the score given by the *i*-th taster for the *j*-th cheese at level k of each factor (year or variety), μ represents their overall mean, $Year_{k}$ is the effect of the *k*-th year ($Year_{2019}=0\;\left[Reference\right]$), $Variety_{k}$ is the effect of the *k*-th variety ($Variety_{pasteur.semi}=0\;\left[Reference\right]$), $Taster_{i}$ and $Cheese_{j}$ are the random effects corresponding to the *i*-th taster and the *j*-th cheese, while $e_{i,j,k}$ is the variability not explained by the remaining effects. It is assumed that random variables $Taster_{i}$ are independent and distributed $N\left(0,\sigma_{T}\right)$, the $Cheese_{j}$ variables are independent and distributed $N\left(0,\sigma_{ch}\right)$ and the error $e_{i,j,k}$ variables are independent and distributed $N\left(0,\sigma_{e}\right)$. The model was estimated by the maximum likelihood Method.

***M3***. ***For each “assessment criteria” year and variety.*** When only a set of permanent tasters and two varieties were considered, the interactions between both factors did not show any statistical significance. Consequently, the data were analysed using the following mixed model:$$y_{i,j,k,l}=\mu +Year_{k}+Variety_{l}+Taster_{i}+Cheese_{j}+e_{i,j,k,l}$$

This model is similar to M1 and M2 but includes the two fixed factors together.

Year–variety interactions were tested using the likelihood ratio test.

#### 2.1.2. Agreement Among Tasters According to Cheese Variety and Year

The agreements in the final scores given by the tasting subsets were determined by the intraclass correlation (IC) coefficient [35], which was estimated using 95% confidence intervals. The IC coefficient measures the agreement of the values of a variable observed by a panel of evaluators—tasters, in this case. Values that come very close to unity indicate a good agreement and those close to zero denote a bad agreement. To conduct this study, the five tasters who evaluated all the *mixed-coagulants* and *vegetable coagulant* cheese varieties in all four years of the study were selected.

Statistical significance was set at p<0.05. Data were analysed by version 4.2.1 of R [36].

## 3. Results and Discussion

### 3.1. Effect of Year and Cheese Variety on the Final Score

Table 2 summarises the frequency distribution according to cheese variety and evaluation year. Given the effect of the interaction between factors “Year” and “Variety” on the “Final score”, the “year” effect was estimated on the final score separately for each variety.

Figure 3 shows how the final scores evolved during the study period for each cheese variety. The *p*-values corresponding to the effect of Year are summarised in Table 3, and as can be seen, five varieties were significantly impacted (*p* < 0.05) depending on the considered tasting year.

As indicated in Figure 3 and Table 3, the *pasteurised mature* cheese [30] obtained the stablest score estimation (*p* = 0.991). The cheese made with *vegetable coagulant* obtained some of the best score estimations, especially in 2020 and 2021. The lowest scores corresponded to the changes in score of the *cow’s milk semi-mature* and *mature* cheeses, and also to the *pasteurised semi-mature* cheeses.

Table 3 shows that the *mixed milk* cheeses (*semi-mature* and *mature*) and the *mixed*-*coagulants* cheeses presented significant differences among the study years. This is probably due to the lack of homogeneity in cheesemakers’ criteria when choosing proportions to make these mixtures, which may influence the disparity in sensorial characteristics between cheeses of the same variety. A similar significance was also observed for the *soft* cheeses (*p* = 0.008). In some cases, the cause could have been related to the different ripening times that the cheeses in these categories were submitted to (from 7 to 34 days for *soft* cheeses and 35–105 days for *semi-mature* cheeses [30]) because, during the ripening time, cheese sensorial characters can considerably vary with aroma formation increasing and secondary *flavours* appearing [37].

The fat content of cheeses made from non-standardized milk varies depending on factors such as the breed of the animal and its diet [38]. For *pasteurised* cheeses, heat treatment also plays an important role [39], and cheese organoleptic characteristics remain more stable because they are made via standard industrialised processes, which cannot be directly compared to artisanal cheese-making processes. In addition, using raw milk confers cheeses more distinctive sensorial characteristics than those made with pasteurised milk [39].

The effects of each variety on the final score were estimated separately for all four study years (Figure 4) (one MM per year). Very significant differences in varieties appeared between the first two study years (*p* < 0.001) (Table 4).

#### Effect on the Final Score Only Considering the Five Permanent Tasters and Two Cheese Varieties

The estimation of the model corresponding to the five permanent tasters and two cheese varieties is summarized in Figure 5.

Figure 5 shows the predicted final scores by variety and across the observation period, as estimated by the M3 model. The curves demonstrate statistically significant differences (*p* = 0.012) with higher scores awarded to cheeses made with the vegetable coagulant. Table 5 displays the differences (95% CI) in annual predictions of final scores, with a notable significant difference observed only between 2019 and 2021.

### 3.2. Effect of Year and Cheese Variety on the Organoleptic Markers

The same analyses were performed for each marker variable. Figure 6 shows how the evolution of scores for each variable for every cheese variety.

Figure 3 and Figure 6 might suggest that there are no stable patterns in tasters’ scores because the evolution of scores barely remains stable throughout the four studied years. When comparing the effects of varieties on the final score in each year (Figure 4), patterns varied over the years, which means that tasters do not follow any standard pattern when making evaluations.

The final scores obtained from tastings revealed some results by clearly showing that the best-valued cheeses were of the *vegetable coagulant* variety, whose good scores remained stable throughout the four study years (Figure 3 and Figure 4). This finding agrees with that reported in the study by Rincón et al. (2016) [26], in which the cheeses made with *goat’s milk* and *vegetable coagulant* stood out for the intensity of their *flavour* and aroma. The finding also agrees with what the tasters in this study determined (see Figure 6).

The analysis of the effects of varieties on the organoleptic markers for each year is summarised in Figure 7 (*odour*), Figure 8 (*texture*), Figure 9 (*flavour*) and Figure 10 (*overall impression*).

If these graphs are analysed, it can be seen that in all graphs there was a greater heterogeneity in the first years and an evolution towards homogeneity, especially in the year 2022 for *texture* (*p* = 0.155) (Figure 8) and *flavour* (*p* = 0.623) (Figure 9), is observed. The *cow’s milk mature* cheese stands out in 2020 because it presents bigger discrepancies than the other varieties for the organoleptic markers *odour* (Figure 7) and *flavour* (Figure 9). For *overall impression* (Figure 10), the same trend is noted (Figure 6). This means that tasters were capable of finding significant differences among all the evaluated cheese varieties for all the organoleptic variables in all the studied years, except in 2022, and as previously mentioned, for *texture* and *flavour*.

As also shown (Figure 6), the greatest heterogeneity existed among the cheese varieties within the range of scores obtained for the variable *flavour*, which, in turn, had the strongest influence on the final score because of its weighting (6 units). Logically, an observation can be made when comparing Figure 3 and Figure 6 that a similar evolution is noted for each cheese variety for the final score and all the organoleptic markers. The only discrepancy with this correspondence was in the case of the evolution of the *goat’s milk semi-mature* cheeses: in 2022, although the final score and most of the sensory variables decreased, *texture* increased.

#### Effect on the Organoleptic Markers Considering Only the Five Permanent Tasters and Two Cheese Varieties

Figure 11 illustrates the predictions estimated by the M3 model for each organoleptic marker by cheese variety over the observation period. Across all markers, cheeses produced with the vegetable coagulant consistently outperformed those made with mixed coagulants, showing statistically significant superiority.

### 3.3. Agreement Between Tasters According to Cheese Variety and Year

The evolution of the agreement among the five permanent tasters for the final scores over the four study years, evaluated by the IC coefficient, for the *mixed-coagulants* and *vegetable coagulant* varieties (the two varieties sampled by all these tasters) is shown in Table 6. For the *mixed-coagulants* variety, the IC was not statistically significant in the first two study years but was in 2021 (*p* = 0.026) and 2022 (*p* = 0.021). Likewise, for the *vegetable coagulant* variety, the IC was not statistically significant in the first three study years but was in 2022 (*p* = 0.03).

Although this analysis was carried out with only two cheese varieties, Table 6 shows that initially, no agreement was reached for the different tasters’ scores, but this improved in later years, particularly for the *mixed-coagulants* cheeses. This improvement may be because these five permanent tasters’ evaluation skills improved over the years. Throughout the evaluation period, the IC tended to show statistical significance, which provides an idea of the trend of the agreement about the evaluations made by the five permanent tasters.

To observe this agreement, and according to all the evaluated organoleptic marker variables, Table 7 shows the corresponding IC coefficients that were found. For the *mixed-coagulants* variety, the IC was not statistically significant for the first two years (2019 and 2020). For 2021, it was significant only for *texture* (p=0.003) but came close to significance for *flavour* and *overall impression*. In 2022, the IC for this same cheese variety was statistically significant for all the markers except *texture* (p=0.157). For the *vegetable coagulant* variety, the IC was only significant for *flavour* in 2022 (p=0.036). For the other marker variables, the IC coefficient came close to significance for *texture* in 2022 (p=0.091), *flavour* in 2019 (p=0.083) and *overall impression* in 2019 and 2022 (p=0.068 and 0.055, respectively).

The best agreements were reached for the *mixed-coagulants* cheeses in the last 2 years studied (2021 and 2022). However, there was some significance when reaching an agreement about the *flavour* and *overall impression* for *vegetable coagulant* cheeses in 2019, but tasters more readily agreed in 2022.

At this point, it is necessary to emphasise the marked difficulty in standardising the cheeses made in the Canary Islands because most are produced in artisanal cheese-making establishments where each cheese maker applies his or her own preparation technique that, in turn, confers on cheeses a distinctive quality and exclusiveness [21]. Furthermore, their cheese characteristics depend on the time of the year when they are assessed because livestock feed and milk production (in litres of milk terms) are not the same in winter as in summer, although the sensory analysis was always carried out in May and June in the competitions in this study. At this point, it is necessary to highlight that most of the milk-producing livestock are from Canarian breeds and they are fed with seasonal plants, hence the importance of the season in the organoleptic characteristics of the cheeses [19].

In the study by Álvarez et al., 2018 [20], the composition of the goat’s milk cheese produced on La Palma Island (Canary Islands) was affected by livestock feed, which significantly increased the sweetness, bitterness and intensity of the *odour* and *flavour* of those cheeses made with milk from goats fed pastureland vegetation as opposed to those that eat wheat straw. The same applies to the cheeses made with milk from the goats on Fuerteventura Island known as Majorera goats because these cheeses have intenser aromas and *odours* [19]. In the present study, this may be related to the annual precipitation there, where the more the recorded rainfall, the more pastureland vegetation there is. Thus, cheese organoleptic quality is also better. When considering this rainfall frequency factor and when more rainfall was recorded, the next year (2019) [40], cheeses obtained higher estimated scores. So, a relation might exist between both factors.

This present study stresses that the cheeses from areas where animals eat pastureland vegetation (*sheep*, *cow* and *vegetable coagulant*) obtained better scores for the *mature* variety than those from where the predominating animal feed is forage (*goats*).

Despite the importance of tasters in sensory analyses, no studies were found that have analysed either the data of sensory evaluations or the evolution of tasters’ scores over time. In fact, Teixeira et al., 2014 [41] point out that no studies are available that have analysed how tasters’ individual actions affect a tasting panel’s final decision. Nonetheless, some works have studied the action of a set of tasters of olive oil [42]. In that study, expert tasters made the evaluation and, to obtain homogeneous results, judges’ extreme scores were ruled out and the homogeneity of the variances among them was studied. Repetitiveness, reproducibility, standard error and confidence intervals (CIs) lowered as the number of judges and replicas increased. Although lower CIs were caused by more repetitions, this was not always justified by tasters’ better work.

Drake (2007) [43] indicated that it is very important to select a tasting panel that is replicable, and whose tasters know and have a well-defined vocabulary when it is used during evaluations. The evaluations made herein were not carried out according to certain sensorial profile descriptors for each cheese variety that had been previously agreed on by tasters [33,44]. Hence, some discrepancies appeared among them. Our tastings according to the UNE 87027:1998 Standard [44] were of the partial sensorial profile kind since only some attributes were evaluated. To properly define the product’s profile, between 12 and 18 judges must be recruited [44], and in our case, a baseline of 26 tasters was taken into account. Furthermore, tasters should be regularly trained [44,45]. Those included in the present study received notions about the functioning and performance of tasting before it commenced. Later, although all our tasters gathered to share their impressions during the evaluation, this did not seem to be sufficient given the discrepancy that appeared among the obtained results. Training and defining the sensorial profile of cheeses and their descriptors are of fundamental importance to achieving the unification of tasters’ criteria [45]. In addition, every time training is given, it would be interesting to present its results to tasters [45] so they know the areas in which they must improve.

To properly make an evaluation of cheeses, the right order in which to evaluate varieties has to be established [45]. During our evaluations, the order followed to evaluate cheeses went from the least *flavour*-intense (*soft cheeses*) to the most *flavour*-intense (*semi-mature and mature*).

Furthermore, each cheese sample should be tasted up to three times to be able to, thus, verify the reliability of the results [44].

So, to perform a sampling of the characteristics taken into account during popular competitions, the profile of tasters can vary and include chefs, influencers, professionals from the sector, or specialised journalists, but they must always have had specific training to participate. With the World Cheese Awards, a panel made up of three non-expert judges performs an initial preselection. Then, another tasting is carried out, but this time with 16 expert tasters to determine which cheeses must be awarded [14]. In our case, the make-up of the participating panel can be justified if it improves and retrains.

Such events are held to not only award the primary sector but to also advertise cheeses to, thus, create an area’s own gastronomic identity and to increasingly promote eating zero-mile products. The importance of these products is such (i.e., their exclusiveness, their limited production, they derive from autochthonous breeds) that cheese routes even exist. These routes act as a tourism advert for visitors and can contribute to economic development [46] in the areas where cheese-making establishments are located, which are highly rural.

## 4. Conclusions

The dynamics of scores awarded in officially organised public cheese-sampling competitions examined in this study reveal the variability in taster evaluations according to tasting year and cheese variety. This work observed that stable patterns in taster evaluations were absent, probably because of a lack of training or the impact of factors that depend on each cheese variety, such as production method, climate, or ripening time, which may also vary from year to year, and even within the same cheese-making establishment. To enhance these dynamics, it is recommended that a validated and referenced analytical method related to the standardization of the number of samples, location, analytical setting, and other variables of interest be established. Nonetheless, tasters were generally capable of clearly distinguishing among the competing cheese varieties by showing preferences for the *vegetable coagulant* and the *semi-mature* cheeses made of both *sheep* and *goat’s* milk, which were perhaps due to their idiosyncrasy and consumption habits.

Although tasters’ assessment criteria tended to be homogeneous towards the end of the study period, relying on some tasters to always participate in consecutive years and sampling several replicas of the same cheese varieties would be desirable to verify this agreement. The study conducted with the five permanent tasters demonstrated a lack of factor interaction, showing greater consistency in their scores. This regularity made it possible to significantly distinguish between years and cheese varieties with the highest and lowest ratings in the official competitions. Thus, to obtain a suitable panel to perform reproducible and repetitive tastings, taking proper training actions and previously defining the sensorial profile of cheese varieties are recommended to correctly identify product-related descriptors, and to also test the correspondence between the sensorial evaluations and the results of the analysed cheese physico-chemical variables.

The diversity of professional profiles of tasting panellists is desirable because, assuming their correct training, this makes these competitions the perfect marketing tool to make a local product visible by assigning cheeses a name and quality and promoting their sale.

## Figures and Tables

**Figure 1 foods-13-03769-f001:**
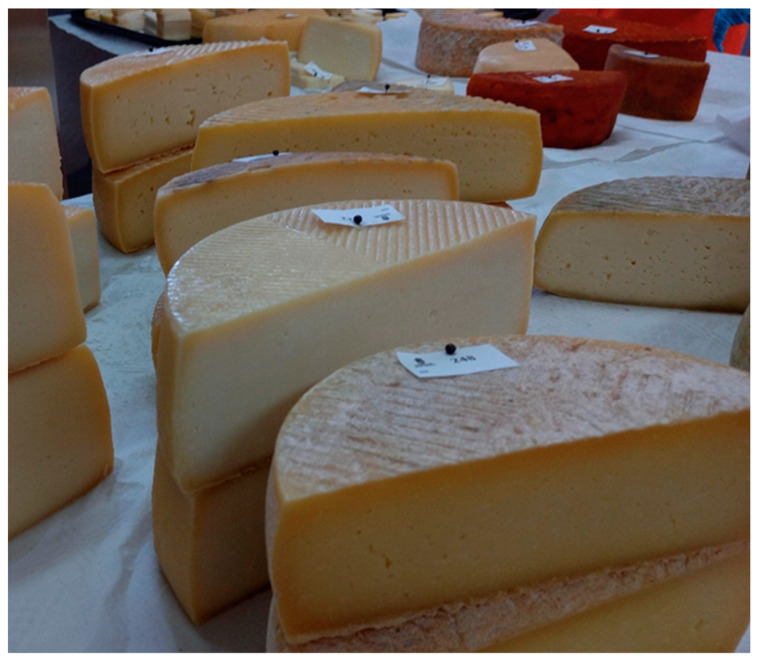
Competing cheeses assigned a random three-digit number.

**Figure 2 foods-13-03769-f002:**
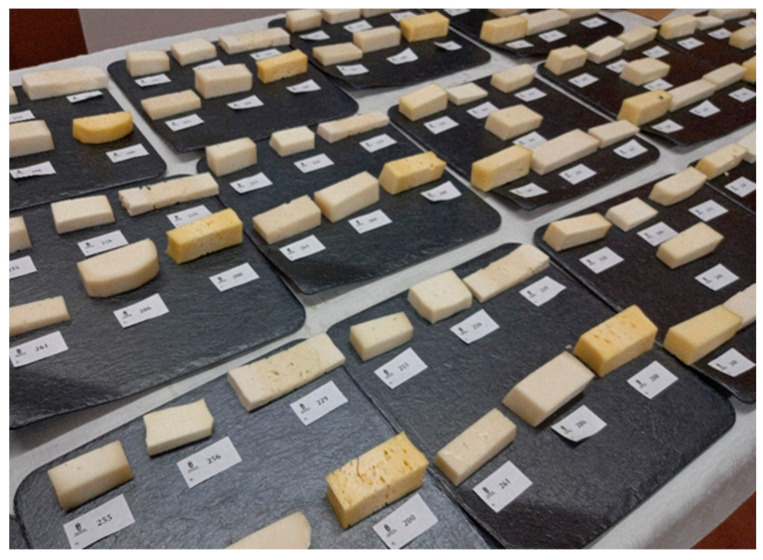
Samples of cheeses prepared for tasting.

**Figure 3 foods-13-03769-f003:**
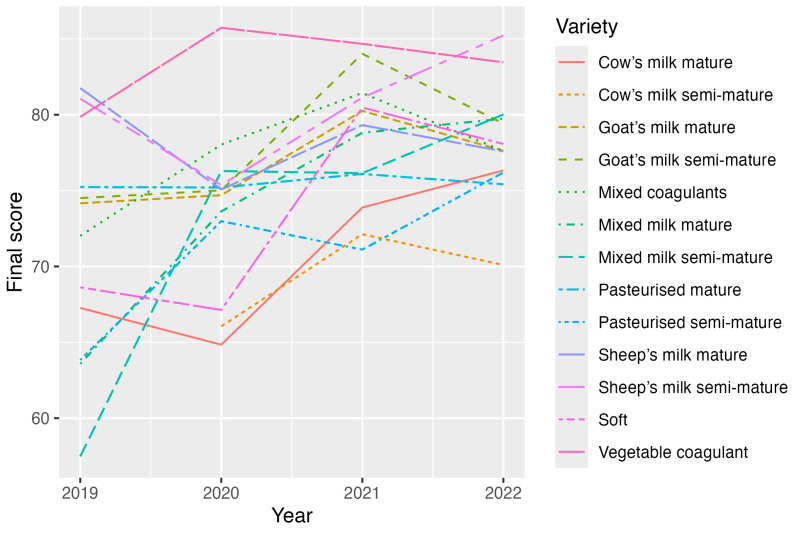
Final score estimates obtained using a separate mixed model for each variety (models of type M1). The graphs clearly show the existence of an interaction between year and cheese variety.

**Figure 4 foods-13-03769-f004:**
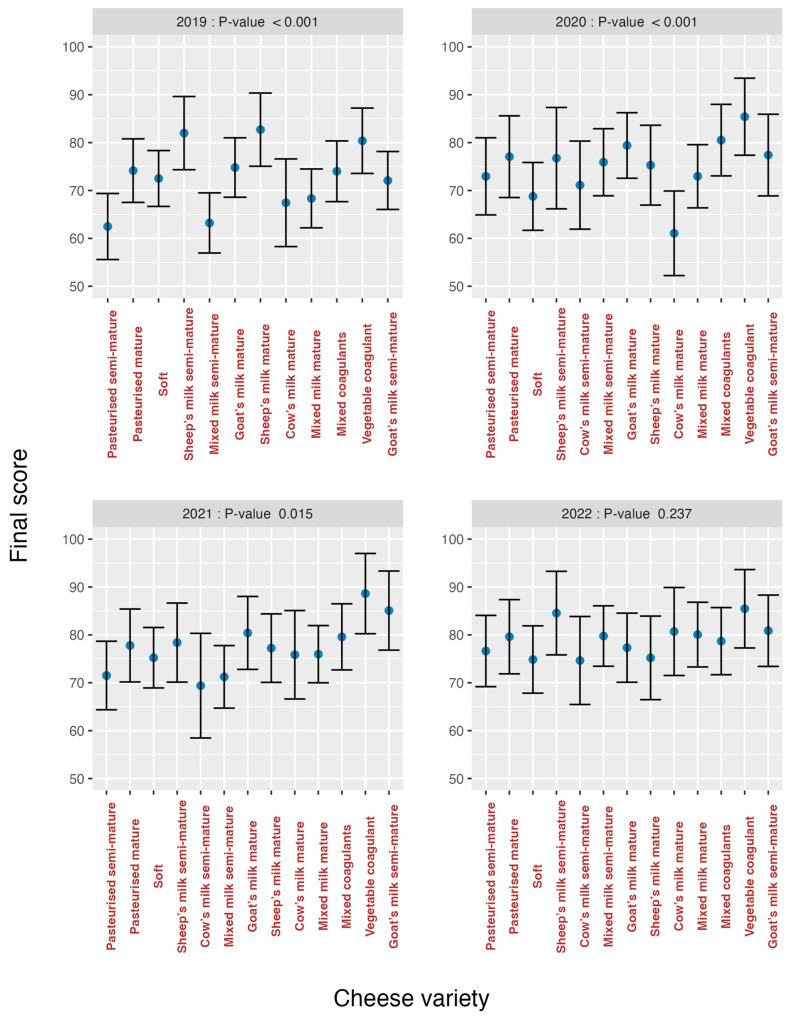
Cheese variety effects on the total score according to the years of study. These graphs confirm the interaction between year and variety.

**Figure 5 foods-13-03769-f005:**
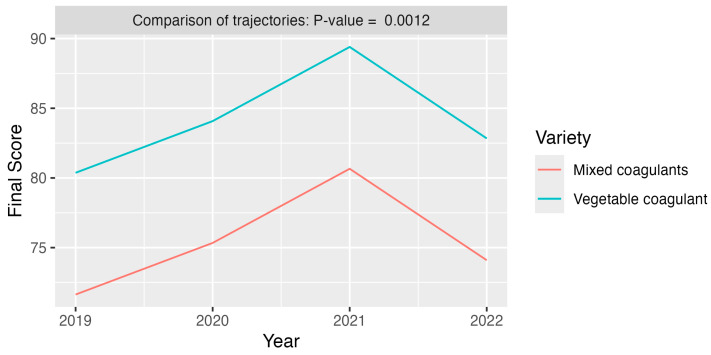
Evolution of the final score according to the factors of year and variety. Since the year–variety interaction did not show statistical significance, we adopted the mixed model of type M3 (no year–variety interaction). The differences in final scores between years are shown in Table 5.

**Figure 6 foods-13-03769-f006:**
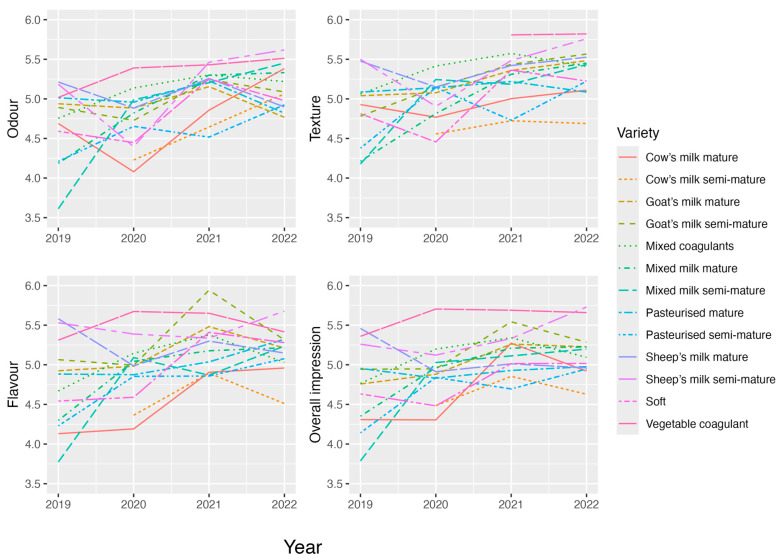
Evolutions of organoleptic markers according to cheese variety.

**Figure 7 foods-13-03769-f007:**
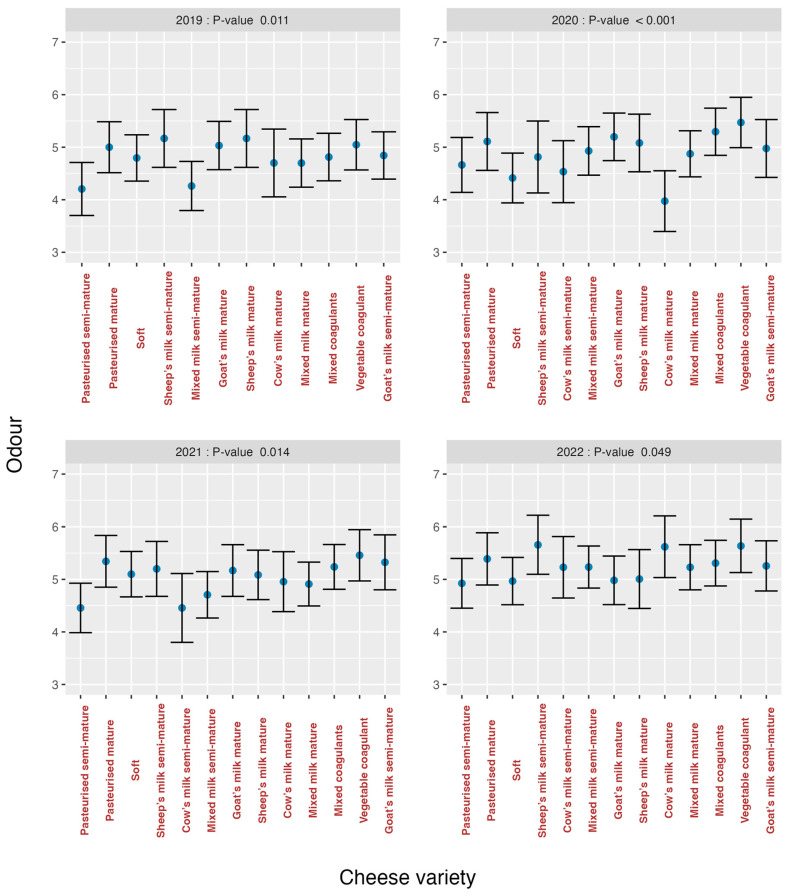
Effect of cheese variety on *odour* according to study year.

**Figure 8 foods-13-03769-f008:**
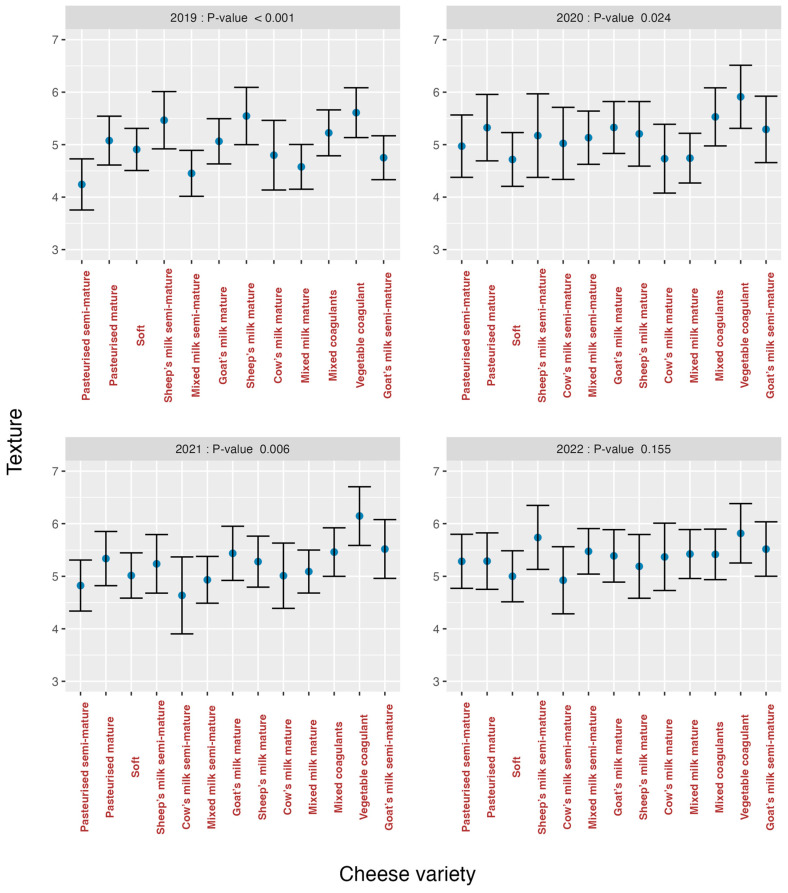
Effect of Cheese Variety on *texture* according to study year.

**Figure 9 foods-13-03769-f009:**
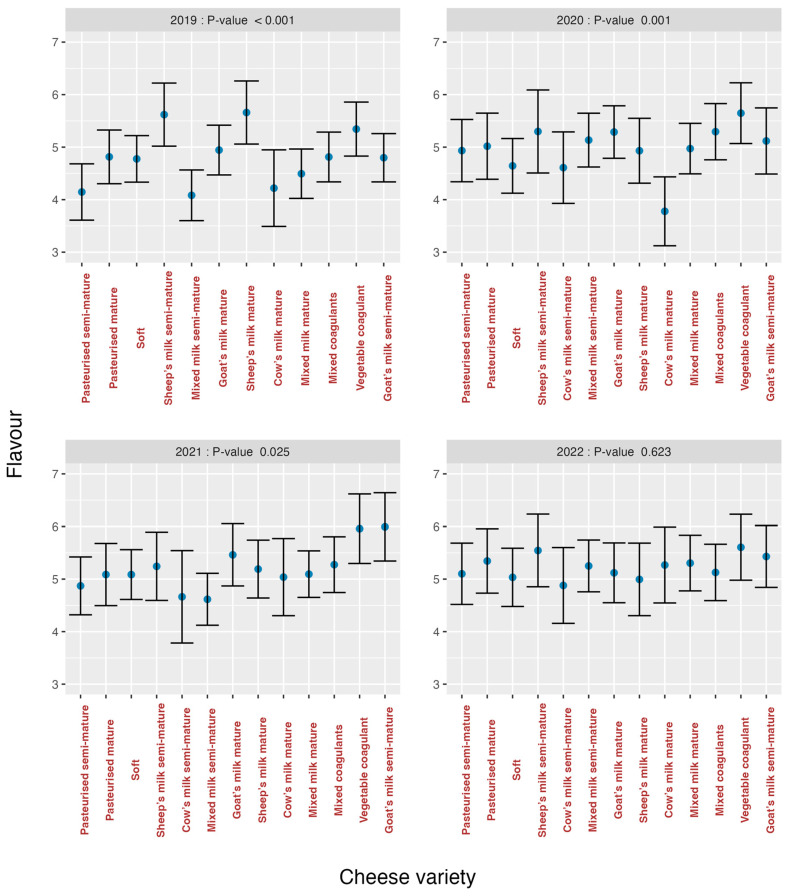
Effect of cheese variety on *flavour* according to study year.

**Figure 10 foods-13-03769-f010:**
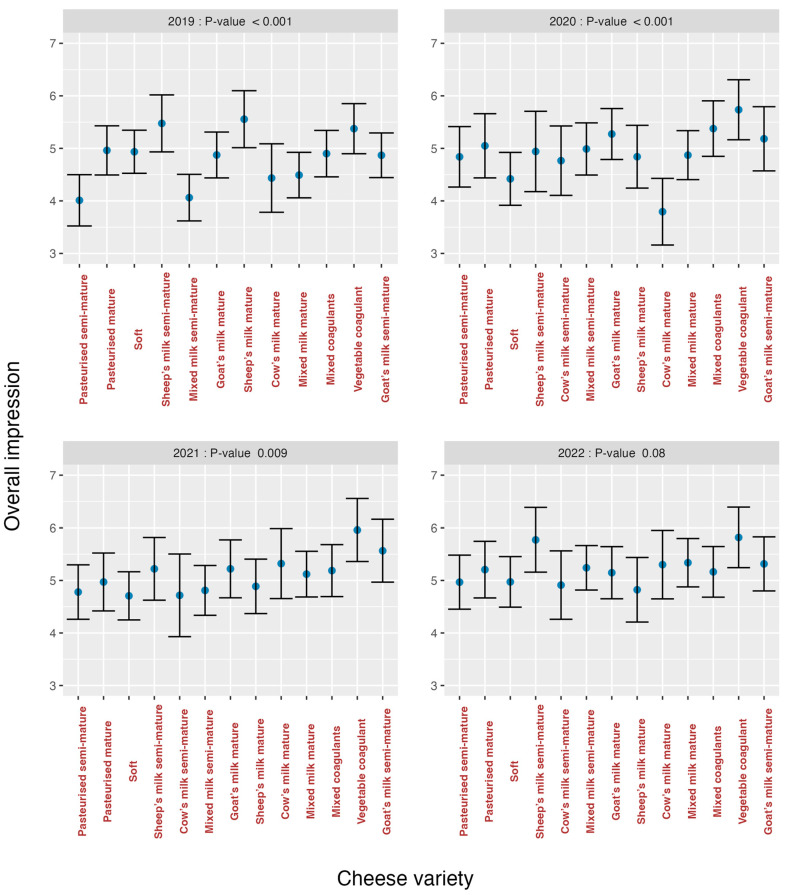
Effect of cheese variety on *overall impression* according to study year.

**Figure 11 foods-13-03769-f011:**
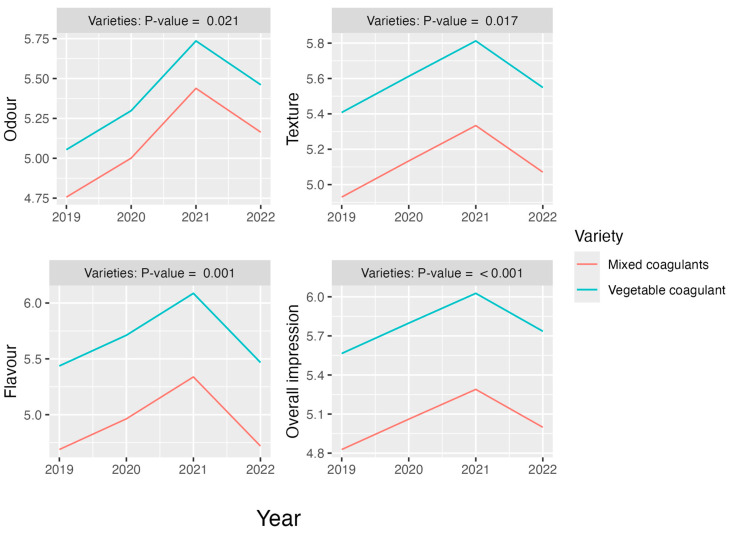
Permanent tasters: evolution of organoleptic markers according to cheese variety. Variety–year interactions did not show statistical significance for any of the four markers. Thus, data were fitted using M3. Annual changes only showed significant differences for odour (between the years 2019 and 2021, *p* = 0.001; between 2020 and 2021, *p* = 0.074).

**Table 1 foods-13-03769-t001:** The cheese varieties evaluated each year, and number of samples and tasters for each variety.

Cheese Varieties	2019		2020		2021		2022	
	No. Samples	No. Tasters	No. Samples	No. Tasters	No. Samples	No. Tasters	No. Samples	No. Tasters
*Pasteurised semi-mature*	7	5	6	6	6	7	6	6
*Pasteurised mature*	7	6	5	6	5	8	6	5
*Soft*	14	5	11	5	9	8	9	5
*Sheep’s milk semi-mature*	5	5	3	5	4	8	4	5
*Cow’s milk semi-mature*	0	0	4	4	2	8	3	6
*Mixed-milk semi-mature*	10	5	10	6	8	8	13	6
*Goat’s milk mature*	9	6	11	6	5	8	8	5
*Sheep’s milk mature*	5	5	6	5	6	8	4	5
*Cow’s milk mature*	3	5	5	5	3	8	3	6
*Mixed-milk mature*	11	5	13	6	11	8	9	6
*Mixed coagulants*	5	16	5	17	5	16	5	11
*Vegetable coagulant*	4	16	4	17	3	16	3	11
*Goat’s milk semi-mature*	10	6	5	6	4	8	7	5
*Total samples*	90		88		71		80	

**Table 2 foods-13-03769-t002:** Tastings by year and cheese variety.

Cheese Variety	2019 N = 575	2020 N = 602	2021 N = 632	2022 N = 482
*Pasteurised semi-mature*	35 (6.1)	36 (6.0)	48 (7.6)	36 (7.5)
*Pasteurised mature*	42 (7.3)	30 (5.0)	40 (6.3)	30 (6.2)
*Soft*	70 (12.2)	55 (9.1)	72 (11.4)	45 (9.3)
*Sheep’s milk semi-mature*	25 (4.3)	15 (2.5)	32 (5.1)	20 (4.1)
*Cow’s milk semi-mature*	0 (0.0)	24 (4.0)	16 (2.5)	18 (3.7)
*Mixed milk semi-mature*	50 (8.7)	60 (10.0)	64 (10.1)	78 (16.2)
*Goat’s milk mature*	54 (9.4)	66 (11.0)	40 (6.3)	40 (8.3)
*Sheep’s milk mature*	25 (4.3)	30 (5.0)	48 (7.6)	20 (4.1)
*Cow’s milk mature*	15 (2.6)	25 (4.2)	24 (3.8)	18 (3.7)
*Mixed milk mature*	55 (9.6)	78 (13.0)	88 (13.9)	54 (11.2)
*Mixed coagulants*	80 (13.9)	85 (14.1)	80 (12.7)	55 (11.4)
*Vegetable coagulant*	64 (11.1)	68 (11.3)	48 (7.6)	33 (6.8)
*Goat’s milk semi-mature*	60 (10.4)	30 (5.0)	32 (5.1)	35 (7.3)

**Table 3 foods-13-03769-t003:** Mixed models of type M1 (n = 13) for the “*final score*” in each cheese variety.

Cheese Variety	Number of Observations in the Model	*p*-Value *
*Pasteurised semi-mature*	155	0.021
*Pasteurised mature*	142	0.991
*Soft*	242	0.008
*Sheep’s milk semi-mature*	92	0.31
*Cow’s milk semi-mature*	58	0.701
*Mixed milk semi-mature*	252	<0.001
*Goat’s milk mature*	200	0.396
*Sheep’s milk mature*	123	0.745
*Cow’s milk mature*	82	0.122
*Mixed milk mature*	275	<0.001
*Mixed coagulants*	300	0.037
*Vegetable coagulant*	213	0.502
*Goat’s milk semi-mature*	157	0.189

(*) Likelihood ratio test for the fixed effect of year.

**Table 4 foods-13-03769-t004:** Mixed models of type M2 (n = 4) for the effects of cheese varieties on the total score per evaluation year.

Year	Number Observations by Model	*p*-Value *
2019	575	<0.001
2020	602	<0.001
2021	632	0.015
2022	482	0.237

(*) Likelihood ratio test for the fixed effect of variety.

**Table 5 foods-13-03769-t005:** Differences in the final scores predicted by M3: data are row-year minus column-year (95% CI). The only significant difference occurs between 2019 and 2021, as the confidence interval does not contain zero.

	2019		
2020	3.71 (−4.76; 12.2)	2020	
2021	**9.03 (0.79; 17.3)**	5.32 (−2.44; 13.1)	2021
2022	2.47 (−6.54; 11.5)	−1.24 (−10.2; 7.7)	−6.56 (−15.6; 2.48)

Differences in bold indicate statistical significance (confidence intervals do not contain zero).

**Table 6 foods-13-03769-t006:** Intraclass correlation coefficients according to cheese variety and year.

Cheese Variety	Year	Intraclass Correlation (IC)	*p*-Value *
*Mixed coagulants*			
	2019	−0.051	0.565
	2020	−0.196	0.946
	2021	**0.333**	**0.026**
	2022	**0.348**	**0.021**
*Vegetable coagulant*			
	2019	0.075	0.277
	2020	−0.121	0.713
	2021	0.164	0.18
	2022	**0.43**	**0.03**

(*) $H_{0}:IC=0$. Bold indicate statistical significance.

**Table 7 foods-13-03769-t007:** The intraclass-correlation (IC) for each organoleptic marker according to cheese variety and year.

		Cheese Variety			
		Mixed Coagulants		Vegetable Coagulant	
Marker	Year	IC	*p*-Value	IC	*p*-Value
** *Odour* **	2019	−0.09	0.677	−0.224	0.968
	2020	−0.207	0.964	−0.201	0.92
	2021	−0.156	0.857	0.082	0.274
	2022	**0.318**	**0.03**	0.246	0.113
** *Texture* **	2019	0.258	0.057	0.09	0.254
	2020	0.053	0.311	−0.214	0.948
	2021	**0.498**	**0.003**	0.429	0.03
	2022	0.147	0.157	**0.28**	**0.091**
** *Flavour* **	2019	−0.076	0.636	0.25	0.083
	2020	−0.138	0.81	0.047	0.327
	2021	**0.263**	**0.055**	0.113	0.235
	2022	**0.333**	**0.025**	**0.407**	**0.036**
** *Overall impression* **	2019	−0.098	0.7	**0.274**	**0.068**
	2020	−0.128	0.783	−0.189	0.891
	2021	**0.239**	**0.07**	0.167	0.178
	2022	**0.366**	**0.017**	**0.353**	**0.055**

IC: Intraclass correlation. Bold indicate statistical significance.

## Data Availability

The original contributions presented in the study are included in the article; further inquiries can be directed to the corresponding author.

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
