# Peer review of "Taste Panellists’ Evaluations in Official Cheese Competitions: Analysis for Improvement Proposals"

_foods, 2024, doi:10.3390/foods13233769_

Round 1
Reviewer 1 Report
Comments and Suggestions for Authors
The manuscript is well written, but some minor and systematic mistakes must be corrected. According to the authors' instructions, the citation in the main text needs to be written with numbers, not the authors' full surnames. Throughout the text, there are a huge number of positions with "Error! Reference source not found.. ". In Table 1. abbreviations need to be written using full terms, or explanations need to be done in the footnote of the table for abbreviations. In Table 2. I suggest writing in the footnote: *Data are frequencies (%) within a year.
Page 8, line 254: Rennet is a term used for animal-origin enzymes (from the stomach). I suggest using the term coagulant.
Figure 6. and Figure 7. Instead of variety, I suggest Cheese Variety.
Page 16, line 350: appears, where?
Page 16, Line 358: Text is missing.
Page 16, Lines 365-366: Text is missing.
Page 17, Line 401: The statement - "more rainfall was recorded" needs to be confirmed by presenting statistical data from the local weather record office.
List of references: The first reference is absent in the main text.
References number 3 and 6 have the same number 5492. Please, check them.
Reference listed under number 8: Who published this study? Please, add.
Reference listed under number 12: Latin names of organisms need to be written using italics.
Reference listed under number 18: please, correct Cordero-Bueno to Cordero-Bueso.
Reference listed under number 39: please delete one dot after the word reliability.

Author Response
Reviewer 1:
First and foremost, I would like to thank your deep review of the manuscript. and the improvement suggestions made. They have undoubtely served to improve the quality and value of the paper.
The manuscript is well written, but some minor and systematic mistakes must be corrected. According to the authors' instructions, the citation in the main text needs to be written with numbers, not the authors' full surnames. Throughout the text, there are a huge number of positions with "Error! Reference source not found.. ". In Table 1. abbreviations need to be written using full terms, or explanations need to be done in the footnote of the table for abbreviations. In Table 2. I suggest writing in the footnote: *Data are frequencies (%) within a year.
- The number of position Error, has been deleted in all the manuscript. We think that it is a mistake of the Office Program between different software versions.
- It is the same mistake in the Table 1 abbreviations, It has now been written correctly, following your recommendations.
- Footnote, Data are frequencies (%) within a year has been changed, following your recommendations. (Page 7, line 25)
Page 8, line 254: Rennet is a term used for animal-origin enzymes (from the stomach). I suggest using the term coagulant.
- We have changed rennet by vegetable coagulant in all manuscript, such as you suggested. (Page 3, line 100-104)
Figure 6. and Figure 7. Instead of variety, I suggest Cheese Variety.
- Cheese Variety has been added to the figures nº 4, 7, 8, 9, 10, such as you suggested. (Page 9, line 287)
Page 16, line 350: appears, where?
- The sentence had been moved one line under, to line 351
Page 16, Line 358: Text is missing (Page 17, line 383).
- The sentence had been moved one line under, to line 359 (Page 17, line 390)
Page 16, Lines 365-366: Text is missing.
- The sentence had been moved one line under, to line 367 (Page 17)
Page 17, Line 401: The statement - "more rainfall was recorded" needs to be confirmed by presenting statistical data from the local weather record office.
- A new reference has been added, nº 40. We have checked it by State Meteorological Agency - AEMET - Spanish Government. (Pag 18, Line 425)
List of references: The first reference is absent in the main text.
- These references have been deleted following your recommendation (Page 20, Line 521)
References number 3 and 6 have the same number 5492. Please, check them.
- Both references nº 1 and 3, were published by AENOR in different years. Moreover, nº1 is a modification of nº3 about Sensorial Vocabulary.
Reference listed under number 8: Who published this study? Please, add.
- This reference has been completed: Albi, M.A.; Gutiérrez, F. Study of the Precision of an Analytical Taste Panel for Sensory Evaluation of Virgin Olive Oil. Establishment of Criteria for the Elimination of Abnormal Results. J Sci Food Agric 1991, 54, 255. (Page 22, line 608-609)
Reference listed under number 12: Latin names of organisms need to be written using italics.
- The scientific name has been changed using italics by your recommendation (Page 21, file 560)
Reference listed under number 18: please, correct Cordero-Bueno to Cordero-Bueso.
- Cordero-Bueso has been added to the references, such as you suggested (Page 21, Line 525)
Reference listed under number 39: please delete one dot after the word reliability.
- It has been deleted following your recommendation (Page 22, Line 594)

Reviewer 2 Report
Comments and Suggestions for Authors
General comments: The manuscript is of high quality and demonstrates a high level of innovation. Among its strengths, I can highlight the thorough detailing of the materials and methods section, the robust statistical support for the results, and the in-depth and comprehensive discussion. I would also like to emphasize the importance of the manuscript in not only covering the research area but also extending the impact of the work beyond the scientific and academic scope. With that said, I recommend the acceptance of the manuscript with minor revisions.
Line 55 - Include a scientific and referenced definition of what cheeses are, and provide a simple global market fact to emphasize the importance and globalization of this product's consumption.
Line 73 - Also mention DOP cheeses.
Line 88 - The paragraph is great and essential, but since you already mentioned ripening in the previous paragraph, start the next one without the term "also."
Line 90 - Mention that these changes result from physical alterations, such as moisture loss and osmotic balance, but are primarily due to biochemical changes like proteolysis and lipolysis, mainly caused by microorganisms from the starter culture or secondary and adjunct cultures, which originate from the raw material or the environment or even by the residual rennet (primary proteolisys).
Line 126 - I would say that the importance and contribution of the study are also relevant to the academic and scientific fields, as it presents a practical execution of sensory methodologies, which are relevant to the market, and employs differentiated methodologies and scopes.
Line 158 - To avoid repeating the word "cheese," use "product" or "sample.
Line 244 - I didn’t quite understand the paragraph due to the "error reference not found" messages. If the p-value mentioned in the paragraph refers to Table 2, I couldn’t find the indicated significant difference in the table.
Line 456 - I would like to suggest indicating that one of the factors that may have caused a lack of stability could be the absence of a validated and referenced analytical method. This method would include a more standardized procedure regarding the number of samples, location, analysis environment, and other relevant factors.
Author Response
Reviewer 2.
General comments: The manuscript is of high quality and demonstrates a high level of innovation. Among its strengths, I can highlight the thorough detailing of the materials and methods section, the robust statistical support for the results, and the in-depth and comprehensive discussion. I would also like to emphasize the importance of the manuscript in not only covering the research area but also extending the impact of the work beyond the scientific and academic scope. With that said, I recommend the acceptance of the manuscript with minor revisions.
- We are deeply thanked by your commentaries about this work and we think that your recommendations could provide significant improvements to the work.
Line 55 - Include a scientific and referenced definition of what cheeses are and provide a simple global market fact to emphasize the importance and globalization of this product's consumption.
- Page 2, Line 53: As you suggested, we have added a new sentence and reference about cheese, Fox and McSweeney (2017).
Line 73 - Also mention DOP cheeses.
- As you suggested, We have added, that are usually locally distinguished or Protected Designation of Origin (PDO)
Line 88 - The paragraph is great and essential, but since you already mentioned ripening in the previous paragraph, start the next one without the term "also."
- As you recommended, “Also” has been added to the beginning of paragraph.
Line 90 - Mention that these changes result from physical alterations, such as moisture loss and osmotic balance, but are primarily due to biochemical changes like proteolysis and lipolysis, mainly caused by microorganisms from the starter culture or secondary and adjunct cultures, which originate from the raw material or the environment or even by the residual rennet (primary proteolisys).
- As you recommended, this paragraph has been rewritten (Pages 3, Line 92-97).
Line 126 - I would say that the importance and contribution of study are also relevant to the academic and scientific fields, as it presents a practical execution of sensory methodologies, which are relevant to the market, and employs differentiated methodologies and scopes.
- As you suggested for standing out the objectives of this work, we have added this paragraph “. In addition, it contributes to the academic and scientific fields, since it helps to validate the functional use of sensory analysis methodology, which is market-relevant and examines the specific practices applied in these official Competitions” (Page3, Lines 128-131)
Line 158 - To avoid repeating the word "cheese," use "product" or "sample.
- As you recommended, “product” has been added (Page 4, Line 165)
Line 244 - I didn’t quite understand the paragraph due to the "error reference not found" messages. If the p-value mentioned in the paragraph refers to Table 2, I couldn’t find the indicated significant difference in the table.
- “The reference source not found” has been deleted. Table 2 has not got p-value, only frequencies of data, nonetheless the Table 3 include p-value about cheese variety. (Page 8, Line 253).
Statistical significance was set at (Line 246)
Line 456 - I would like to suggest indicating that one of the factors that may have caused a lack of stability could be the absence of a validated and referenced analytical method. This method would include a more standardized procedure regarding the number of samples, location, analysis environment, and other relevant factors.
- As you suggested, we have added a new paragraph for including your recommendation about the factors that may have caused a lack stability, “To enhance these dynamics, it is recommended to establish a validated and referenced analytical method related to the standardization of the number of samples, location, analytical setting, and other variables of interest” (Page 19, Lines 482-485)

Reviewer 3 Report
Comments and Suggestions for Authors
Scientific research is usually based on formulating a hypothesis and aims, designing an experiment, and then obtaining results and evaluating them. A somewhat different approach than usual has been applied to this manuscript. The authors obtained an extensive set of results from a four-year sensory evaluation of cheeses and tried to statistically evaluate these results and draw conclusions. Unfortunately, the results obtained are influenced by several variable factors that are mentioned in the text.
I therefore appreciate the large amount of data obtained, the thorough and professional statistical evaluation. On the other hand, the approach "we have a lot of numbers from 4 years of sensory assessment from different samples and different evaluators, we will try to calculate something from them and publish it" does not seem completely scientific to me. So, I'll leave it to the editor to consider this opinion of mine.
Moreover, I would like to point out to some specific comments that I would like to be considered:
L46: Did the "trained panellists" meet the conditions according to the ISO standard?
L54-57: They might be some important information about the worldwide cheese consumption on the IDF webpages.
There is an error in citation of the tables and figures in the text. See lines 149, 159, 161 etc., where there is the text “Error! Reference source not found.”
All tables should be more self-explanatory. Please, specify more the titles, explain, why some variables are underlined, why some values have different colours, and put the units into the table (not in the notes bellow the table).
Numbers of observations in the model given in Table 2 is repeated in Table 3.
L270: Fat contend in cheeses from non-standardised milk depends mainly on the breed of the animal. This is usually stronger factor than heat treatment.
L361: It might be useful to evaluate separately the four-year results obtained only from these five permanent assessors (as the effect of assessor is minimised).
Your panellists have also been on different levels of qualification. How was this factor considered in the evaluation of your results?
L390: Seasonal effect is important, but also the breed of animal plays the role.
Author Response
Reviewer 3
Scientific research is usually based on formulating a hypothesis and aims, designing an experiment, and then obtaining results and evaluating them. A somewhat different approach than usual has been applied to this manuscript. The authors obtained an extensive set of results from a four-year sensory evaluation of cheeses and tried to statistically evaluate these results and draw conclusions. Unfortunately, the results obtained are influenced by several variable factors that are mentioned in the text.
I therefore appreciate the large amount of data obtained, the thorough and professional statistical evaluation. On the other hand, the approach "we have a lot of numbers from 4 years of sensory assessment from different samples and different evaluators, we will try to calculate something from them and publish it" does not seem completely scientific to me. So, I'll leave it to the editor to consider this opinion of mine.
First of all, we would like to thank you for the opportunity that your comments have given us to significantly improve the content of the article. On the other hand, the work has been carried out with scientific rigor based on the data provided by the official entity organizing the competitions, concerned about the convenience of improving the methodology followed and minimizing biases in the development of these events to award prizes that are truly deserved. Finally, we consider that the work also has an economic-socio-cultural interest and for the maintenance of traditions, with useful results in future local events and extrapolated to any contest of similar characteristics anywhere in the world.
Moreover, I would like to point out to some specific comments that I would like to be considered:
L46: Did the "trained panellists" meet the conditions according to the ISO standard?
- Really the tasted panellists can be considered from a point of view overall, like laboratory judges or trained pannellists (I am an expert taste panellist and always have participated in this panel and other taste competition), but the Official Cheese Tasting also include other people with less experience. Thus, the conditions could be considered seem to ISO Standard.
L54-57: They might be some important information about the worldwide cheese consumption on the IDF webpages.
- As you recommended, new information about cheese consumption has been added. What is more, “the consumption of this product was about 9.2 million metric tons in 2023 at the European Union, meanwhile in China, it was about 409 thousand metric tons in 2019” [8]. (Page 2, Line 57-59).
There is an error in citation of the tables and figures in the text. See lines 149, 159, 161 etc., where there is the text “Error! Reference source not found”
- As you suggested, all mistakes as “Error! Reference source not found” have been delated. We think that it is a mistake with the Office Program between different software versions.
All tables should be more self-explanatory. Please, specify more the titles, explain, why some variables are underlined, why some values have different colours, and put the units into the table (not in the notes bellow the table).
- As you recommended, all Tables have been reviewed and more information has been added. All Tables are explained in the new text.
Numbers of observations in the model given in Table 2 is repeated in Table 3.
- It is correct, but the p value of number of observations in the model were included in the Table 3 and the Table 2 there are only frequencies (%) and samples number. We considered that in this way, it would be clearer for the reader.
L270: Fat contend in cheeses from non-standardised milk depends mainly on the breed of the animal. This is usually stronger factor than heat treatment.
- As you suggested, we have added a new commentary about the fat content of cheese and new reference, 38. “The fat content of cheese made from non-standardized milk varies depending on fac-tors such as the breed of the animal and its diet [38]. For pasteurised cheeses, heat treatment also plays an important role” (Page 9, File 279).
L361: It might be useful to evaluate separately the four-year results obtained only from these five permanent assessors (as the effect of assessor is minimised).
- As you suggested, we have created a new section entitled “3.1.1. Effect on the final score only with the permanent tasters and two cheese varieties” Also, we have included a new Figure (5) and new Table (5) with a new paragraph for explaining this new section about the five permanent assessors. (Page 10, Lines 294-310)
Your panellists have also been on different levels of qualification. How was this factor considered in the evaluation of your results?
- Please, let me make a little commentary about level of qualification of taster panelist. It is true that this would be an interesting factor to include in the study, but it could not be considered since there was no access to this information, which was reserved by the Official Organizing Entity of the Contests, perhaps to respect anonymity and privacy. right to protection of personal data of tasters. Although the choice of participants is random and they are always excluded as “external” personnel, what we do have information about is that all of them would fall into the categories of trained or laboratory judges and in this way, we have included this information in the new version of the text.
L390: Seasonal effect is important, but also the breed of animal plays the role.
- As you recommended, we have added a new paragraph to explain the breed of animal effect. “At this point it is necessary to highlight that most of the livestock are from Canarian breeds and they are feed with the seasonal plants, hence the importance of the season in the organoleptic characteristics of the cheeses [19]” (Pages 18, Lines 413-415).

Round 2
Reviewer 1 Report
Comments and Suggestions for Authors
The authors respected all suggestions. According to my opinion, the manuscript is ready for publishing.
Reviewer 3 Report
Comments and Suggestions for Authors The authors responded to all my questions and comments. Comments were incorporated into the manuscript. I recommend accepting the manuscript for publication.